# Side-binding proteins modulate actin filament dynamics

**Alvaro H Crevenna[1,2]\*, Marcelino Arciniega[3,4], Aurélie Dupont[1,5,6], Naoko Mizuno[7], Kaja Kowalska[2], Oliver F Lange[4,8,9], Roland Wedlich-Söldner[2,10], Don C Lamb[1,5,6]**

[1]Physical Chemistry, Department of Chemistry and Center for Nanoscience, Ludwig-Maximilians-Universität München, Munich, Germany; [2]Cellular Dynamics and Cell Patterning, Max Planck Institute of Biochemistry, Martinsried, Germany; [3]Max Planck Institute of Biochemistry, Martinsried, Germany; [4]Department of Chemistry, Technische Universität München, Garching, Germany; [5]NanoSystems Initiative Munich, Ludwig-Maximilians-Universität München, Munich, Germany; [6]Center for Integrated Protein Science Munich, Ludwig-Maximilians-Universität München, Munich, Germany; [7]Cellular and Membrane Trafficking, Max Planck Institute of Biochemistry, Martinsried, Germany; [8]Biomolecular NMR and Munich Center for Integrated Protein Science, Technische Universität München, Garching, Germany; [9]Institute of Structural Biology, Helmholtz Zentrum München, Neuherberg, Germany; [10]Institute of Cell Dynamics and Imaging, and Cells-in-Motion Cluster of Excellence (EXC 1003 – CiM), University of Münster, Münster, Germany

**\*For correspondence:** alvaro.crevenna@cup.uni-muenchen.de

**Competing interests:** The authors declare that no competing interests exist.

**Abstract** Actin filament dynamics govern many key physiological processes from cell motility to tissue morphogenesis. A central feature of actin dynamics is the capacity of filaments to polymerize and depolymerize at their ends in response to cellular conditions. It is currently thought that filament kinetics can be described by a single rate constant for each end. In this study, using direct visualization of single actin filament elongation, we show that actin polymerization kinetics at both filament ends are strongly influenced by the binding of proteins to the lateral filament surface. We also show that the pointed-end has a non-elongating state that dominates the observed filament kinetic asymmetry. Estimates of flexibility as well as effects on fragmentation and growth suggest that the observed kinetic diversity arises from structural alteration. Tuning elongation kinetics by exploiting the malleability of the filament structure may be a ubiquitous mechanism to generate a rich variety of cellular actin dynamics.

## Introduction

Central cellular processes such as cell migration, cytokinesis, endocytosis, and mechanosensation depend critically on actin-based force generation and actin filament turnover (**Pollard and Borisy, 2003**; **Lecuit et al., 2011**). The molecular basis of actin filament turnover derives from the association and dissociation of monomers from each filament end and depends on the nucleotide (ATP, ADP · Pi, or ADP) bound to the actin monomer (**Pollard, 1986**). The filament is kinetically asymmetric, where one end (called the barbed-end) is observed to grow an order of magnitude faster than the other end (the pointed-end) (**Pollard, 1986**). In addition, the critical concentration for polymerization is different for the two ends. The origin of the asymmetry is not fully understood. Measurements of filament

**eLife digest** Actin is one of the most abundant proteins in cells. It forms networks of filaments that provide structural support and generate the forces needed for cell movement, division, and many other processes in cells.

Filaments of actin continuously change in length as actin molecules are added or removed at the ends. One end of an actin filament—called the barbed-end—grows much faster than the other, known as the pointed-end. Many other proteins also help the actin filaments to form. Some of these proteins bind to the ends of the filaments, where they directly control the growth of the filaments. Other proteins bind along the length of the filaments, but how these 'side-binding' proteins influence the growth of filaments is not clear.

In this study, Crevenna et al. used a technique called 'total internal reflection fluorescence (TIRF) microscopy' to study how several side-binding proteins affect the growth of actin filaments in an artificial system. The growth of the barbed-ends was strongly influenced by which side-binding protein was interacting with the filament. For example, the barbed-end grew rapidly when a protein called VASP was present but grew more slowly in the presence of the protein α-actinin. Although the growth at the pointed-end was generally slow and sporadic, the side-binding proteins also had noticeable effects.

Crevenna et al. found that when the side-binding proteins were present at low levels, filament growth was similar for all proteins studied. It was only when the proteins were present at higher levels that the growth of the actin filaments was altered depending on the specific side-binding protein present. One side-binding protein called α-actinin also altered the shape of the actin filament so that when it was present at high levels, the filaments curved in a particular direction. Together, these results suggest that the growth, structure, and flexibility of actin filaments can be strongly influenced by the various proteins that bind along the length of the filaments.

The next challenges are to understand the precise details of how these side-binding proteins are able to alter the growth and shape of actin and investigate how they influence other processes that control the structure of actin networks in cells.

elongation as a function of solution viscosity (*Drenckhahn and Pollard, 1986*) and particle-analysis from cryo-electron microscopy (*Narita et al., 2011*) suggest the existence of a non-elongating state at the pointed-end. Although growth pauses have been previously observed during filament elongation measured using total internal reflection fluorescence (TIRF) microscopy (*Kuhn and Pollard, 2005*; *Fujiwara et al., 2007*), these pauses were attributed to artifacts and were not characterized further. The dynamics of the pointed-end plays an important role in both the origin of the differences in critical concentration observed at the two ends in the presence of ATP (*Pollard, 1986*; *Fujiwara et al., 2007*); and in filament treadmilling, where, barbed-end growth and pointed-end shrinking occur simultaneously (*Bugyi and Carlier, 2010*). Thus, we have focused on performing an accurate and detailed analysis of both barbed-end and pointed-end dynamics using TIRF microscopy.

In cells, a large number of proteins interact with actin filaments, either at the ends or with the lattice. End-binding proteins regulate actin dynamics by limiting elongation (at the barbed-end) or serving as anchor points (for the pointed-end). Side-binding proteins, on the other hand, are much more diverse encompassing myosin motors, cross-linkers or bundlers as well as severing proteins. The interaction of the actin filament with a particular subset of proteins defines the molecular composition, architecture, and overall turnover of sub-cellular arrays such as stress fibers and filopodia. Some of these arrays are tightly packed (*Jasnin et al., 2013*) and dynamics of the filaments are influenced by the local environment. The mechanisms of how some proteins are recruited to these structures while others are excluded are a subject of intense research (*Cai et al., 2008*; *Hansen et al., 2013*). Although the overall filament dynamics have been thought to be sensitive to the concentration of the side-binding protein (*Breitsprecher et al., 2009*), it is not understood how and to what extent side-binding proteins alter filament kinetics, structure, and flexibility.

In this study, we used TIRF microscopy to study the effect of side-binding proteins on the dynamics of actin filament growth in vitro. We chose three cross-linking proteins and one motor protein to represent the large variety of interacting proteins and used them to tether filaments directly to the

surface of a glass slide for visualization. We used the chemically inactivated myosin II motor protein (NEM-myosin) as it is the standard choice for this type of assay (*Kuhn and Pollard, 2005*). The filamin protein (*Kueh et al., 2008*) was used, which is an important player in cellular mechanosensing that is evolutionary-conserved (*Razinia et al., 2012*), as its use as a tether has recently generated some debate (*Mullins, 2012*; *Niedermayer et al., 2012*). Additionally, we selected α-actinin, a molecule that, together with myosin II, forms stress fibers (*Langanger et al., 1986*), and VASP, a protein that localizes to areas of dynamic actin reorganization such as filopodia and the lamelipodium (*Rottner et al., 1999*). By carrying out these assays with several proteins that bind to the side of actin filaments, we were able to explore the possible range of modulation available to actin filament dynamics and delineate intrinsic filament properties.

## Results

### Kinetic modulation at the barbed-end

Fluorescently labeled actin was used to visualize the growth of actin filaments (*Figure 1A–B*) using TIRF microscopy. In this technique, single actin filaments are tethered to a glass surface via a side-binding protein and their growth and/or shrinkage is monitored in real time (*Figure 1A–B*). From each frame, the filament is extracted and a kymograph is constructed (*Figure 1—figure supplement 1*). The position of each end of the filament was then determined by fitting an error function (*Demchouk et al., 2011*) to each line of the kymograph (see 'Materials and methods' and *Figure 1—figure supplement 1* for details). This end-detection method provides a more accurate determination of the filament length and thereby a more reliable estimate of the instantaneous elongation velocity compared to methodologies used previously (*Figure 1—figure supplement 1*).

The single-filament elongation experiments showed that the barbed-end grew at a constant velocity with occasional pauses for all constructs measured while the barbed-end elongation velocity varied depending on the particular side-binding protein used (*Figure 1C,D*). The elongation velocity '$E$' at the barbed-end was the fastest with VASP and the slowest with α-actinin (*Figure 1C*). By varying the free actin concentration from 0.3 to 2 μM, we estimated the barbed-end association and dissociation rates, $k_{on}$ and $k_{off}$, respectively (*Figure 1D*), using only the periods of elongation (i.e., E > 1.5 sub·s$^{-1}$, referred to hereafter as the 'kinetically active' phases). Compared to the previously reported value of 11.6 sub·μM$^{-1}$·s$^{-1}$ for actin only in the absence of tethering proteins (*Pollard, 1986*), we found a higher value of $k_{on}$ in the presence of VASP, a similar value for actin alone and for NEM-myosin, and lower values when α-actinin or filamin were used (*Figure 1D* and *Table 1*). Extrapolating the elongation velocity as a function of actin concentration to zero actin provides an estimate of the dissociation rate, $k_{off}$, of ATP-actin at the barbed-end (*Table 1*). In the presence of filamin, $k_{off}$ is indistinguishable from zero, whereas in the presence of VASP, $k_{off}$ increased compared to the value in the presence of NEM-myosin (1.6 ± 0.5 s$^{-1}$). The estimated $k_{off}$ we measured in the presence of NEM-myosin was in agreement with the previously reported value of 1.4 s$^{-1}$ (*Pollard, 1986*), whereas in the presence of α-actinin, $k_{off}$ was lower than 1.4 s$^{-1}$. The ratio of inferred dissociation rates to the calculated association rate (i.e., $k_{off}/k_{on}$) is the critical concentration at which polymerization will occur and has been estimated to be ~150 nM for the barbed-end (*Pollard, 1986*). We find a similar value (~0.2 μM) for filaments elongated in the presence of VASP, α-actinin and NEM-myosin, but close to zero for filamin. The use of VASP induced the largest change in the measured kinetics. In contrast to the other three proteins measured, the kinetics were enhanced. VASP is known to act as a polymerase at the barbed-end by delivering subunits to the growing end (*Hansen and Mullins, 2010*; *Breitsprecher et al., 2011*). It achieves this function through its two actin-binding domains: a G-actin-binding domain that delivers monomers (via G-actin binding or GAB domain) while remaining attached to the filament via the F-actin-binding or FAB domain (*Breitsprecher et al., 2008, 2011*; *Hansen and Mullins, 2010*). To rule out the polymerase activity of VASP as the cause of enhanced kinetics, we also tested a VASP construct that lacks the GAB domain but retains its capacity to interact with the filament lattice (*Breitsprecher et al., 2008*). We continued to observe fast polymerization using this VASP-ΔGAB protein as a tether (*Figure 1*) in agreement with previous reports (*Breitsprecher et al., 2008*). The measured kinetic rates were 70 ± 30 sub·μM$^{-1}$·s$^{-1}$ for $k_{on}$ and 14 ± 9 s$^{-1}$ for $k_{off}$, about half of those determined using the full-length VASP construct. Therefore, the effect of immobilized VASP on filament kinetics is not only due to recruitment of monomers to the

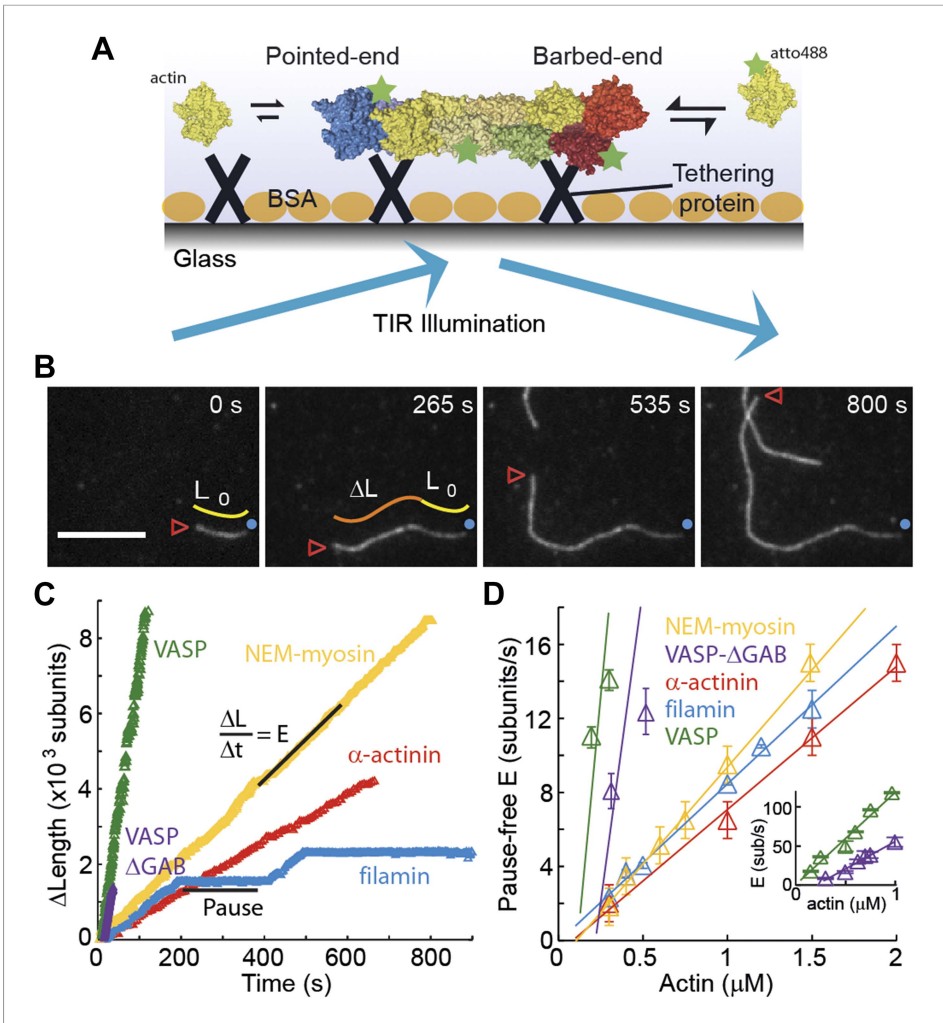

**Figure 1**. The dependence of the barbed-end kinetics on the side-binding protein. (**A**) A schematic of total internal reflection illumination and single actin filament imaging of filaments tethered to a glass surface. Filaments grow from the addition of subunits at either the barbed- or the pointed-end. (**B**) Selected frames from a movie showing the growth of a single actin filament that is tethered to the surface via α-actinin. The barbed-end is marked by a red arrowhead and the pointed-end by a blue dot. The elapsed time interval is given in seconds. Scale bar: 5 μm. $L_0$ and $\Delta L$ are the initial filament length and the change in length, respectively. (**C**) $\Delta L$ as a function of time for single filaments grown on surfaces with different tethering proteins. (**D**) Elongation velocity (E) as a function of actin concentration in solution for different tethering proteins (inset, zoom out of the VASP and VASP ΔGAB values). The elongation velocity was determined from the slope of the graphs of $\Delta L$ vs time in regions where no pauses were observable. Error bars represent s.e.m. (n > 20). Tether density here is ~2000 molecules/μm².
The following figure supplement is available for figure 1:

**Figure supplement 1**. Comparison of algorithms for end-detection and filament growth.

---

growing filament. These results demonstrate that ATP-actin kinetics at the barbed-end are sensitive to the particular side-binding protein interacting with the filament.

## Kinetic modulation at the pointed-end

Pointed-end association and dissociation rates were estimated in the same manner as those for the barbed-end (*Figure 2A*). Both the estimated association rates and dissociation rates varied according to the associated side-binding protein used as a tether (*Figure 2A* and *Table 1*).

**Table 1.** Rate constants of Mg-ATP-actin monomer association and dissociation at both ends of the actin filament in the absence and presence of side-binding proteins

| | End | $k_{on}$ (sub·µM$^{-1}$·s$^{-1}$) | $k_{off}$ (sub·s$^{-1}$)† | $k_{off}/k_{on}$ (µM) | Reference |
|---|---|---|---|---|---|
| actin alone | Barbed | 11.6 ± 1.2 | 1.4 ± 0.8 | 0.12 ± 0.07 | (*Pollard, 1986*) |
| | Pointed | 1.3 ± 0.2 | 0.8 ± 0.3 | 0.6 ± 0.17 | (*Pollard, 1986*) |
| | Barbed | 9.7 ± 2* | 1 ± 0.3 | 0.1 ± 0.04 | this work |
| | Pointed | 2.1 ± 0.8 | 0.8 ± 0.4 | 0.4 ± 0.35 | this work |
| Surface adsorbed | | | | | |
| NEM-myosin | Barbed | 11 ± 1 | 1.6 ± 0.7 | 0.15 ± 0.03 | this work |
| | Pointed | 0.8 ± 0.1 | 0.4 ± 0.1 | 0.5 ± 0.2 | this work |
| *Dd* VASP | Barbed | 120 ± 30 | 1 ± 3 | 0.01 ± 0.03 | this work |
| | Pointed | 48 ± 10 | 0.5 ± 2 | 0.01 ± 0.05 | this work |
| filamin | Barbed | 8.5 ± 1.3 | 0.1 ± 0.4 | 0.012 ± 0.002 | this work |
| | Pointed | 5.3 ± 0.1 | 2.6 ± 0.2 | 0.5 ± 0.04 | this work |
| α-actinin | Barbed | 7.7 ± 1.5 | 0.7 ± 1 | 0.1 ± 0.2 | this work |
| | Pointed | 0.9 ± 0.3 | 0.9 ± 0.3 | 1 ± 1 | this work |
| *Dd* VASP ΔGAB | Barbed | 70 ± 13 | 14 ± 9 | 0.2 ± 0.2 | this work |
| | Pointed | 16 ± 12 | 5 ± 8 | 0.3 ± 0.2 | this work |
| In solution | | | | | |
| *Dd* VASP | Barbed | 126 ± 30 | 43 ± 33 | 0.3 ± 0.2 | this work |
| | Pointed | 12 ± 8 | 3 ± 8 | 0.3 ± 2 | this work |
| filamin | Barbed | 8.6 ± 1.1 | −1.3 ± 2 | 0.0 ± 0.1 | this work |
| | Pointed | 5.5 ± 1.5 | 2.8 ± 1.6 | 0.5 ± 0.4 | this work |
| *Dd* VASP ΔGAB | Barbed | 24 ± 11 | 4 ± 15 | 0.2 ± 1 | this work |
| | Pointed | 3 ± 2.5 | 0.5 ± 4.5 | 0.2 ± 7 | this work |
| *Hs* VASP | Barbed | 24 ± 4 | −3 ± 5 | 0 ± 0.1 | (*Hansen and Mullins, 2010*) |
| | Pointed | Not reported | Not reported | Not reported | (*Hansen and Mullins, 2010*) |

*All reported errors from this work are 95% confidence intervals whereas those of (*Pollard, 1986*) represent SD.
†All reported dissociation constants from this work are inferred from extrapolation of the elongation velocity as a function of actin concentration to zero concentration, data from *Figures 1, 2 and 4*.

The presence of filamin increased the $k_{on}^P$ by a factor of ~5 (from 0.8 in the presence of NEM-myosin to 2.8 sub·µM$^{-1}$·s$^{-1}$). The $k_{on}^P$ for α-actinin was 0.9 sub·µM$^{-1}$·s$^{-1}$, while, when using VASP or VASP ΔGAB, the rate was 44 sub·µM$^{-1}$·s$^{-1}$ and 16 sub·µM$^{-1}$·s$^{-1}$, respectively. On the other hand, the presence of filamin also increased the inferred $k_{off}^P$ by almost an order of magnitude from 0.4 (in the presence of NEM-myosin) to 2.6 s$^{-1}$. The inferred $k_{off}^P$ rates were 0.7 s$^{-1}$, 8 s$^{-1}$, and 5 s$^{-1}$ with α-actinin, VASP, and VASP ΔGAB, respectively (*Table 1*).

Unlike the barbed-end where there were occasional pauses (*Figure 1C*), the pointed-end displayed mostly a kinetically inactive phase or paused state and only grew sporadically (*Figure 2B,C*). Such kinetically inactive phases were observed for all free actin concentrations tested (250 nM–2 µM). Above the pointed-end critical concentration (e.g., using a free actin concentration of 1 µM), we observed a discontinuous (i.e., growth-pause) behavior for all side-binding proteins (*Figure 2B*). In the presence of VASP or filamin, pointed-end elongation was readily observed. Pointed-end elongation was much more difficult to visualize when using NEM-myosin and α-actinin (*Figure 2B*) where elongation occurred for brief periods of time and with slower rates. The elongation velocity during kinetically active phases was influenced strongly by the different tethering proteins used (*Figure 2A*). Elongation velocity followed the order of VASP > VASP ΔGAB > filamin > α-actinin > NEM-myosin (*Figure 2A,B*). On the other hand, at 300 nM free actin monomer concentration, i.e., below the pointed-end critical concentration of ~600 nM (*Pollard, 1986*), we observed barbed-end growth

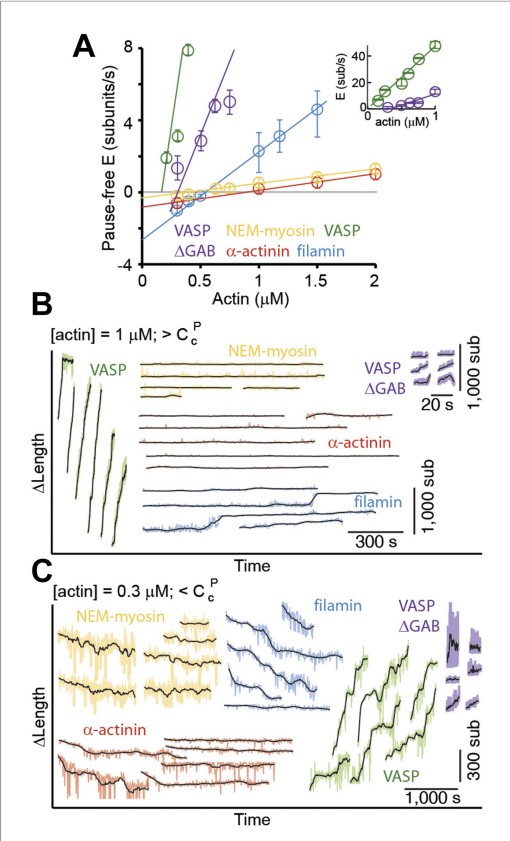

**Figure 2**. Pointed-end elongation and depolymerization kinetics as a function of the associated side-binding protein. (**A**) The elongation velocity (E) is plotted as a function of free actin concentration. Error bars are s.e.m. (n > 20). (**B–C**) A gallery of traces of ΔL as a function of time for pointed-ends observed at (**B**) 1 µM or (**C**) 0.3 µM free actin monomer concentration for the different tethering proteins studied. The raw data are shown in color, and the black solid lines are a running average of 10 data points.

(*Figure 1D*) and pointed-end depolymerization (*Figure 2C*), i.e., treadmilling, in the presence of filamin as a tethering protein (*Figure 2C*). Treadmilling was also present using NEM-myosin and α-actinin, albeit with slower rates, since pointed-end depolymerization establishes the overall treadmilling rate. In contrast to our expectations, there was no shrinkage at the pointed-end below the critical concentration but polymerization in the presence of VASP or VASP ΔGAB (*Figure 2*). These results suggest that side-binding proteins can also determine actin filament pointed-end growth and depolymerization dynamics. Additionally, these results show that observed effects at one end do not necessarily represent effects at both ends. For example, filamin reduces only the dissociation rate (and therefore the critical concentration) at the barbed-end although it alters both the association and dissociation rate at the pointed-end.

## The elongation rate varies with occupancy of the side-binding proteins

Next, we studied how sensitive filament dynamics are to the presence of each of the proteins tested. Therefore, we measured the elongation rates and pausing as a function of the side-binding protein surface density (*Figure 3*). For this, we varied the total protein concentration that was allowed to adsorb to the glass surface, therefore changing the number of tethering proteins that interact with a single filament. We estimated the lattice-binding protein surface density from the protein concentration, the sample volume (∼10 µL) and the surface to which the sample was adsorbed (a flow cell of 5 mm × 20 mm, giving 100 mm²) as done previously (*Howard et al., 1989*; *Crevenna et al., 2008*). All protein in solution was assumed to adsorb on the upper and lower glass surfaces. To achieve consecutive lower tether densities, the total protein concentration was serially diluted. At low tethering protein concentrations, individual filaments swiveled around distinctive attachment points indicating that they are bound to single tethering molecules as observed previously (*Howard et al., 1989*; *Crevenna et al., 2008*). To estimate the density in an alternative manner, we measured the average number of pivot points per micron of filament at the two lowest protein concentrations and divided that by the average area covered during swiveling. Assuming a linear scaling with protein concentration, this estimate results in a lower density (by a factor of 2) compared to those reported in *Figure 3* and throughout the text. Estimated densities ranged from ∼5 up to ∼18,000 molecules·µm⁻², which are equivalent to values between 0.1 and 110 tethers per micron of filament (*Figure 3*).

At low tether densities (5–200 µm⁻² or 0.025 to 1.0 molecules per micron of filament), the dynamics were independent of the tethering protein used. As an example, filamin is shown in *Figure 3A–D*. Elongating actin filaments (at a free actin monomer concentration of 1 µM) showed mostly kinetically active phases (*Figure 3A*), and the elongation velocity distributions were centered around ∼9 subunits·s⁻¹ (*Figure 3* and *Figure 3—figure supplement 1*). At high tether densities (600–18,000 µm⁻²

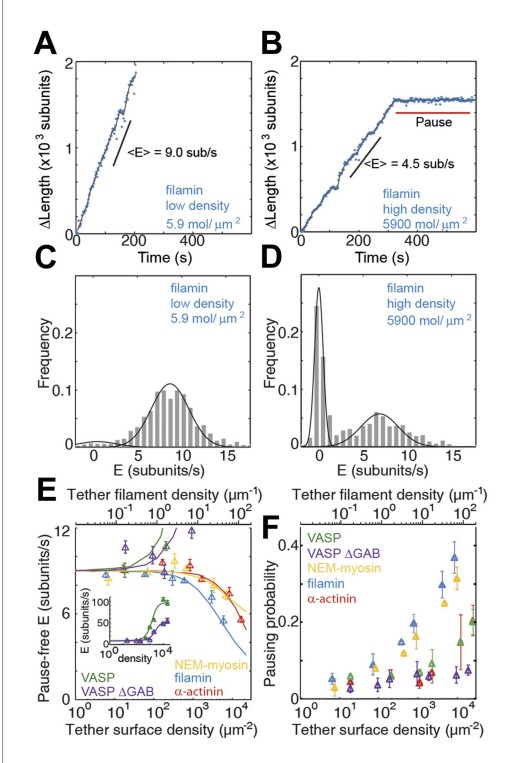

**Figure 3**. Barbed-end actin filament elongation as a function of the surface density of side-binding proteins. (**A–B**) The change in length, $\Delta L$, of actin filaments as a function of time when using filamin as the surface tethering protein at the (**A**) lowest (5.9 molecules/$\mu m^2$ or 0.03 molecules per micron of filament) or (**B**) the highest (5900 molecules/$\mu m^2$ or 35 molecules per micron of filament) density. (**C–D**) Distribution of elongation velocities for filaments using a filamin-coated surface at the (**C**) lowest or (**D**) highest density. Solid lines are fits to Gaussian distributions. The distribution is calculated by binning the instantaneous elongation velocity of more than 20 filaments into 0.75 subunits/s bin size. (**E**) Elongation velocity as a function of tether surface density estimated from the kinetically active phases. The surface density is plotted as number of tethering proteins per unit surface area on the lower axis and the equivalent number of tethering proteins per $\mu m$ of filament on the upper axis. Solid lines are fits to a model where protein binding induces an allosteric effect that persists along the filament over a certain length scale (see 'Materials and methods' for details). (**F**) Pausing probability as a function of surface tether density. Error bars represent s.e.m. ($n > 20$).

The following figure supplements are available for figure 3:

**Figure supplement 1**. Actin filament elongation as a function of the surface density of side-binding proteins.

**Figure supplement 2**. Barbed- and pointed-end actin

*Figure 3. continued on next page*

or 3–110 molecules per micron of filament), each side-binding protein tested generated a particular elongation behavior (*Figure 3E* and *Figure 3—figure supplement 1*). Using filamin, increasing the surface tether density decreased the mean elongation velocity of kinetically active phases (*Figure 3B,D*) and increased the fraction of time the filament spent in a paused state, i.e., the pausing probability '$P_p$' (*Figure 3B,D,F*). In contrast, increasing the VASP or the VASP-$\Delta$GAB density increased the elongation velocity while VASP also increased the $P_p$ (*Figure 3* and *Figure 3—figure supplement 1*). Higher surface concentrations of $\alpha$-actinin or NEM-myosin had also an effect on the elongation velocity (*Figure 3—figure supplement 1*) and, in addition, the density of NEM-myosin had a strong effect on the $P_p$ (*Figure 3F* and *Figure 3—figure supplement 1*).

One possible explanation for these results could be geometric and/or mechanical constrains imposed on the filament by the high density of the surface-immobilized side-binding protein used. To investigate this possibility, we carried out experiments where a very low density of NEM-myosin was used to tether filaments to the surface while a second, side-binding protein was present in solution. We tested the effects of VASP, VASP-$\Delta$GAB, and filamin on filament growth. The influence of all three side-binding proteins on the elongation rate of both the barbed- and pointed-end were similar to what we observed when using them to immobilize the actin filaments to the surface (*Figure 3—figure supplement 2* and *Table 1*). We also performed these experiments with human VASP and the elongation rates measured for both *Dd* VASP and *Hs* VASP agree with previously reported results (*Breitsprecher et al., 2008*; *Hansen and Mullins, 2010*) (*Figure 3—figure supplement 3*). These results suggest that a variety of elongation kinetics can arise from the specific interaction of actin filaments with the particular associated side-binding protein.

## Intrinsic filament dynamics

To further verify that the observed kinetic changes and pauses originate from the particular side-binding protein used as a tether, we investigated the intrinsic properties of filament elongation and controlled for artifacts. Single elongating filaments were measured at the lowest protein surface density possible that still allowed filament visualization. At the lowest $\alpha$-actinin tether density used (5 molecules $\mu m^{-2}$, which corresponds to 1 tether molecule every

*Figure 3. Continued*

filament elongation kinetics.

**Figure supplement 3**. Barbed-end actin filament elongation as a function of side-binding proteins concentration.

**Figure supplement 4**. Schematic of the proposed model.

**Figure supplement 5**. Comparison of the expected behavior using a higher local concentration mechanism with experimental results.

5–10 microns along the filament), the ends swiveled around their tethering site due to Brownian motion and were clearly free of the surface (*Figure 4A*). Under these conditions, barbed-ends showed continuous elongation (*Figure 4B*) while pointed-ends were typically in the kinetically inactive state (only 2 of 50 filaments showed growth or depolymerization, *Figure 4C–D*). For the other tethering proteins, only the paused state was observed on freely swiveling pointed-ends (*Figure 4—figure supplement 1*). Using only the pause-free elongation velocities for each actin concentration tested, we estimated association and dissociation rates (slopes in *Figure 4E*, *Table 1*). Our estimated values for the pause-free elongation kinetics agree well with those previously obtained by EM (*Table 1*), which were measured on the 20–60 s time scale. When we convolute our pointed-end pause-free elongation rate with the pausing probability, our results are comparable to the kinetics estimated by TIRF experiments (*Kuhn and Pollard, 2005*). One possible explanation for this discrepancy is that continuous pointed-end growth occurs at the beginning of filament assembly, which is suggested from our data (*Figure 4C*) and is the time scale on which the EM data was acquired (*Pollard, 1986*). Moreover, the pausing probability, $P_p$, at either end was insensitive to the actin concentration used (*Figure 4F*).

The low density used for these experiments and the observed pauses on freely swiveling actin filaments (pointed-end only) rules out surface effects (*Kuhn and Pollard, 2005*) as the determining cause for the pauses at the ends. Another possible source of pauses is light-induced photo-dimerization. From the work of *Niedermayer et al. (2012)*, it is possible to quantitatively predict the accumulated fraction of filaments where depolymerization has been paused as a consequence of exposure to light (*Figure 4—figure supplement 2*). In contrast with this prediction, we observed all swiveling filament pointed-ends, under depolymerizing conditions, to be in a kinetically inactive state at the beginning of image acquisition (N = 40, *Figure 4—figure supplement 2*). Only in the presence of a medium to high density of tethering proteins did we observe depolyermization of pointed-ends (12 of 55, *Figure 4—figure supplement 2*).

As an additional test to rule out any tether, surface or light-induced effect of the pausing, we used a two-color solution assay to investigate pointed-end growth. Here, a small seed (formed with atto565-labeled actin) was allowed to grow in solution for 15 min in the presence of atto488-labeled monomers, followed by stabilization, dilution, and visualization of the filaments (*Figure 4—figure supplement 3*). At a free actin concentration of 1 μM, the concentration used in solution to allow filament elongation, all pointed-ends are expected to grow at an average rate of ~0.5 sub/s (*Pollard, 1986*). In contrast to this expectation, we observed that only 20% of the seeds grew at the pointed-end (N = 1000, *Figure 4—figure supplement 3*). This percentage is higher than we observe in the surface-based experiments, which could be due to annealing of filaments in solution (*Sept et al., 1999*; *Andrianantoandro et al., 2001*) or due to lack of the tethering protein. What is clear is that the non-elongating or paused state is not due to either surface or light-induced effects. Taken together, these results show that a single rate constant describes filament elongation kinetics from ATP-monomers in the absence of side-binding proteins and that the pointed-end has an intrinsic kinetically inactive state.

## Structural effects of side-binding proteins on filaments

During the course of filament elongation analysis as a function of side-binding protein density on the surface (*Figure 3*), we noticed that filaments appeared more bent as the tether density increased. To quantify this curviness, we estimated an apparent persistence length '$L_p^*$' of individual filaments associated with different side-binding proteins (see 'Materials and methods' for details). The persistence length $L_p$ (*Boal, 2012*) reflects the material properties of the filament, which are related to its structure (*Chu and Voth, 2005*, *2006*; *Pfaendtner et al., 2010*), and has already

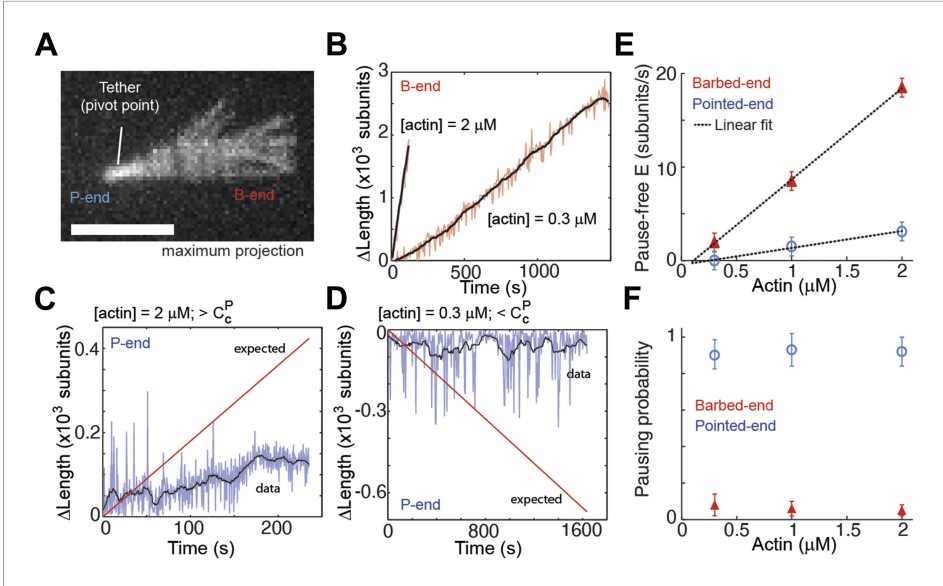

**Figure 4**. Intrinsic filament dynamics. (**A**) A maximum projection image from a movie of an actin filament tethered to a glass surface via a single α-actinin molecule where the tethering position about which the filament swivels is visible as a constriction point. Scale bar: 3 μm. (**B**) Change in length of the barbed-end vs time for individual actin filaments attached to the surface using the lowest tethering protein surface densities at 300 nM and 2 μM concentrations of free actin monomers. (**C–D**) Change in length vs time of the pointed-ends of single, elongating actin filaments using the lowest tethering protein surface densities and either 2 μM (**C**) or 300 nM (**D**) of actin monomers free in solution. The red lines represent the expected elongation behavior based on previously reported rates using NEM-myosin as a tether (*Kuhn and Pollard, 2005*; *Fujiwara et al., 2007*). (**E**) The pause-free elongation velocity (E) plotted as a function of free actin concentration. The lines represent linear fits. Estimated rates are reported in *Table 1*. Error bars are s.e.m. (n > 20). (**F**) Pausing probability as a function of free actin concentration. Error bars represent s.e.m. (n > 20).

The following figure supplements are available for figure 4:

**Figure supplement 1**. Pointed-end pausing on freely swiveling ends for various tethering proteins.

**Figure supplement 2**. The distribution of the time to the first elongation pause at 300 nM free actin concentration.

**Figure supplement 3**. Two-color seeded assay for visualizing pointed-end growth from an actin filament seed.

been shown to be tunable by side-binding proteins (such as myosin or cofilin [*McCullough et al., 2008*; *Murrell and Gardel, 2012*; *Bengtsson et al., 2013*]). At the lowest side-binding protein density (~10 molecules μm⁻² or ~0.1 molecules per filament micron), actin filaments had an $L_p^*$ of ~18 μm and was independent of the associated protein (*Figure 5A*). At the highest densities (~16,000 molecules·μm⁻² or ~100 molecules per filament micron), the presence of NEM-myosin decreased the $L_p^*$ to 4 ± 1 μm while it was reduced to 3 ± 1 μm, 5 ± 1 μm, and 2.2 ± 0.3 μm when using filamin, VASP and α-actinin, respectively (N > 50 for each condition, *Figure 5A*). Estimates for the persistence length of surface adsorbed filaments are consistent with what has been determined for freely fluctuating filaments (*McCullough et al., 2008*; *Graham et al., 2014*). We also tested the mechanical effect of the side binding proteins on the actin filament when the actin filaments were attached to the surface with a low density of NEM-myosin and the side-binding protein was present in solution. Again, we observed a decrease in the persistence length of about 30% for filamin, α-actinin, and NEM-myosin, whereas the effect was about 50% in the presence of VASP or VASP ΔGAB (*Figure 5A*).

Two other interesting phenomena were observed in the presence of side-binding proteins at high densities. First, the presence of filamin increased the spontaneous fragmentation of filaments (20 out of 197 filaments vs less than 1 fragmentation even per 200 filaments) (*Figure 5B*). Second, barbed-end

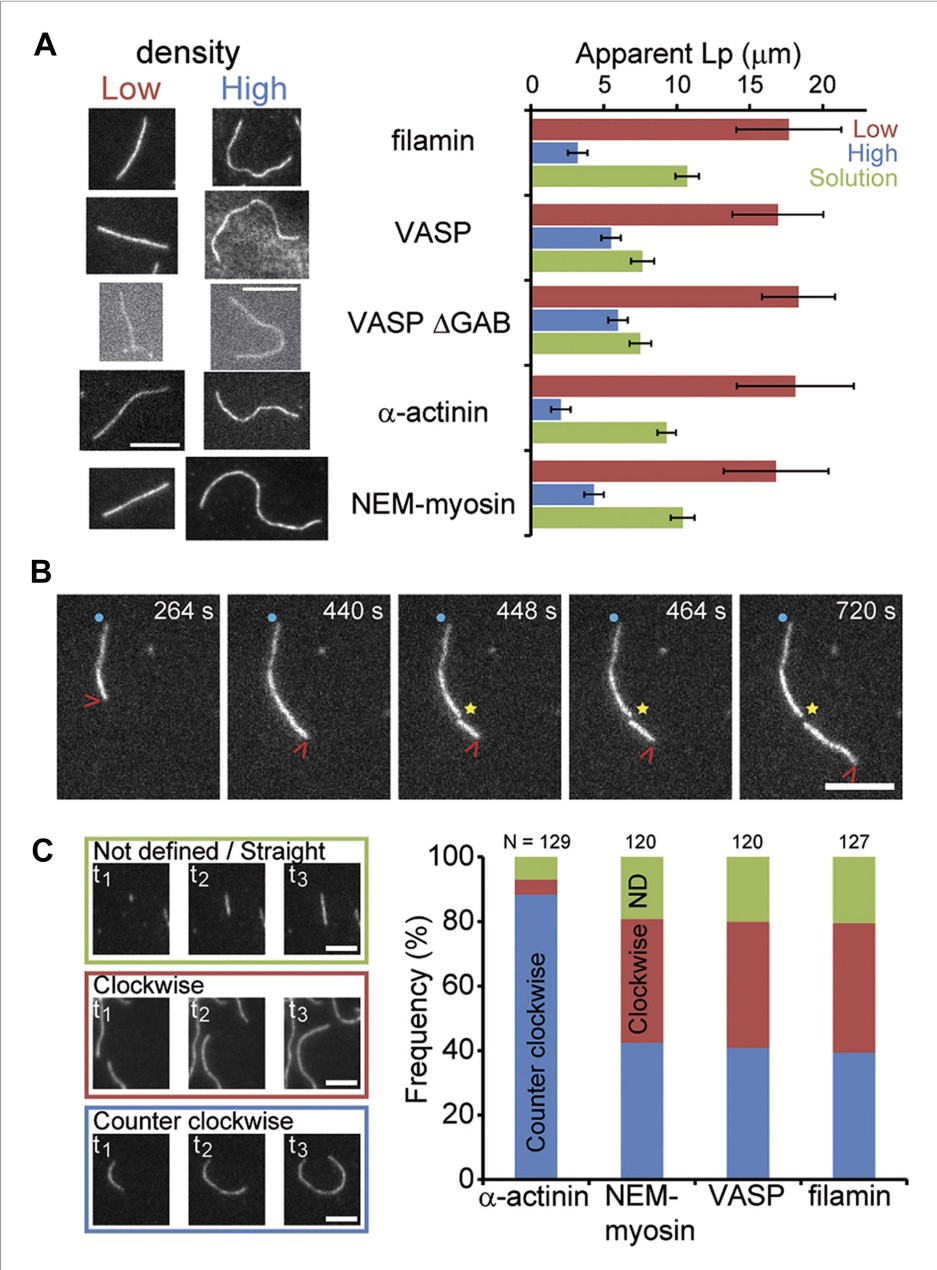

**Figure 5**. Side-binding proteins alter filament structure. (**A**) (left panels) Images of individual filaments attached to the surface using different side-binding proteins at the lowest or highest surface density of tethering protein. Scale bar: 5 µm. (right panels) Estimated apparent persistence length from the angular correlation along the filament contour length at the lowest (red) and the highest (blue) lattice-binding protein densities, and when the protein is present in solution (green). Error bars represent s.e.m. of more than 50 filaments measured per experimental condition. (**B**) Images from a movie of an individual growing actin filament under treadmilling conditions. The barbed-end is marked with a red arrowhead and pointed-end with a blue dot. The filament undergoes a fragmentation event (yellow star) at 488 s and afterwards depolymerizes from its new pointed-end while the newly created barbed-end does not elongate. The free-actin concentration was 400 nM. Time is given in seconds. Scale bar: 5 µm. (**C**) Characterization of the direction of barbed-end filament growth as a function of the tethering protein used (see 'Materials and methods' for details). Examples of each class are shown in the left panels. Scale bars: 3 µm. α-actinin was observed to grow almost exclusively in the counterclockwise direction. No preferred direction of growth is observed for the other side-binding proteins measured.

elongation when tethered with α-actinin had a preference to grow in a counterclockwise direction (*Figure 5C*). This counterclockwise elongation observed when α-actinin was present was independent of the length the filament had when it landed on the surface. These observations suggest an influence of the tethering protein on the structural properties of the filament.

Changes in the structure of the filament by binding proteins are known to be able to propagate over several subunits (*Orlova et al., 1995*). We hypothesize that structural alteration might be the origin of kinetic modulation. To test how well this interpretation could explain our results, we constructed a simple model based on long-range structural alterations to describe elongation as a function of tether density (*Figure 3E*). In our model, the interaction between a growing filament and a tethering protein gives rise to a modified association rate (i.e., $\alpha \cdot k_{on}^0$) at the site of interaction, which then propagates over a certain distance $L_C$ (*Figure 3—figure supplement 4*). From the interaction site, there is a linear decay of the modified association rate, as a linear decay would be expected for the release of torsional stress. Using a Monte Carlo method (see 'Materials and methods' for details), we calculated $\alpha$ and $L_C$ (which are the only free parameters) by comparing the simulated elongation rates to the experimental values and minimizing the $\chi^2$ (*Table 2*). This simple model satisfactorily described our experimental results (*Figure 3E*, solid lines) and provides an estimate for the propagation length. We also considered the possibility that VASP acts via a 'local increase in monomer concentration' similar to *Breitsprecher et al. (2011)* (see 'Materials and methods' for details). This local concentration model did not account for the tether density dependence of elongation velocity (*Figure 3—figure supplement 5*), nor it can account for the effect observed with VASP ΔGAB that does not have the capacity to bind actin monomers.

Our estimate of the propagation length should be considered as a lower limit since additional factors that could potentially influence our assay such as tether unbinding (i.e., which would lower the effective number of interacting side-binding proteins) and/or alternative tether density calculations (see above) would result in longer propagation lengths. The induced effects are local-to-short-ranged (~2-11 monomers) for α-actinin, filamin, and NEM-myosin while they are long-ranged when using VASP and VASP-ΔGAB (160 and 76 monomers, respectively) (*Table 2*). Collectively, these results suggest that the side-binding proteins tested alter the structure of the filament.

## Discussion

We have shown that asymmetry in filament elongation is a consequence of a non-elongating state at the pointed-end and that the general versatility of actin dynamics may be a response to the binding of various proteins. Through accurate measurements of pointed-end association kinetics, we have observed that experiments performed in the absence of tethering proteins and in the presence of VASP or VASP ΔGAB yielded equivalent critical concentrations for both ends (~0.2 μM, *Table 1*). This implies that, under these conditions, detailed balance at equilibrium is fulfilled, i.e., $k_{off}^B / k_{on}^B = k_{off}^P / k_{on}^P$ (*Hill, 1987*). We propose that the existing discrepancy in estimated critical concentrations at both ends (*Fujiwara et al., 2007*) originates from the presence of a previously uncharacterized kinetically inactive or non-elongating state at the pointed-end. This kinetically inactive state is consistent with a non-elongating structural conformation observed by cryo-electron microscopy (*Narita et al., 2011*). The kinetic asymmetry of the pure actin filament may be low (~3) and such non-elongating or closed conformation at the pointed-end would reinforce the effective filament asymmetry. The transition at the pointed-end from the open to the closed state may be coupled to ATP hydrolysis or phosphate release at the terminal

**Table 2**. Results of a Monte Carlo simulation describing the affect of lattice protein binding to the association rate of actin monomer binding to filaments

|  | α | $L_C$ (monomers) |
|---|---|---|
| VASP | 9* (7–10)† | 160* (145–175)† |
| VASP ΔGAB | 5.1 (5.0–5.1) | 76 (74–76) |
| α-actinin | 0.4 (0.4–0.7) | 1 (1–11) |
| Filamin | 0.4 (0.2–0.8) | 11 (1–101) |
| NEM-myosin | 0.7 (0.3–0.9) | 11 (1–201) |

The binding of an actin-binding protein onto the lattice of a filament leads to changes (with magnitude α) in association kinetics that are propagated over a certain characteristic length $L_C$, as a number of monomers.
*The value obtained by minimizing the $\chi^2$.
†The values in parenthesis represent the 68% confidence interval.

subunit. The presence of this open-to-closed transition at the pointed-end would explain why the terminal subunit has an estimated different rate of phosphate release compared to the filament lattice (*Fujiwara et al., 2007*).

Actin filaments in association with any of the five proteins tested displayed a change in elongation velocity, an increase in pausing, and a change in filament flexibility. Therefore, it is possible that these three characteristics have a common origin. For three of these filament-binding proteins (myosin, α-actinin, and filamin), the binding interface to the actin filament is formed by two consecutive monomers along the same strand (*Galkin et al., 2008*, *2010*; *Lorenz and Holmes, 2010*). These side-binding proteins might directly occlude the binding site for the next monomer either partially (reducing the elongation velocity) or completely (giving rise to an elongation pause). Partial distortion of the filament could turn into a defect that propagates along the lattice decreasing the observed filament stiffness and impacting the association rate. In this respect, side-binding proteins could be thought of as allosteric regulators of actin filament kinetics. Indeed, actin filaments are known to be subject to allosteric regulation by other associated proteins (*Egelman and Orlova, 1995*; *Galkin et al., 2012*). In particular, myosin (*Prochniewicz and Thomas, 1997*), cofilin (*Galkin et al., 2001*; *Prochniewicz et al., 2005*), dystrophin (*Prochniewicz et al., 2009*), and utrophin (*Prochniewicz et al., 2009*) are known to induce structural changes in the actin filament. Similar to filamin and α-actinin, dystrophin and utrophin bind actin through calponin-homology (CH) domains (*Galkin et al., 2010*). Moreover, binding to the filament is cooperative for cofilin (*De La Cruz, 2005*), αE-catenin (*Hansen et al., 2013*), and myosin (*Orlova and Egelman, 1997*). The basis for this allosteric regulation could originate from the stabilization of an existing structural state of the filament (*Galkin et al., 2001*), given that the actin filament is structurally polymorphic (*Galkin et al., 2010*). Therefore, it is possible that the observed elongation kinetics and pauses arise from direct modulation of the filament structure. In line with this hypothesis, two other proteins, the actin-binding domain of αE-catenin (*Hansen et al., 2013*) and an N-WASP construct (*Khanduja and Kuhn, 2014*), have recently been shown to alter filament kinetics and one of them, the actin-binding domain of αE-catenin, also influences filament structure (*Hansen et al., 2013*). Although atomically accurate simulations and more high resolution experiments are required to understand the molecular basis of monomer association and dissociation from the filament ends, our results provide evidence that lattice structural changes affect actin filament growth kinetics. The influence of different side-binding proteins on the growth kinetics was found to persist over different length scales. Although we do not currently know the mechanism of this difference, it is interesting to note that actin-binding proteins with globular binding domains (α-actinin, filamin, and NEM-myosin) have short-range affects whereas VASP and VASP ΔGAB, which have an unstructured binding motif, have more long-range affects.

Our experimental approach of using tethers immobilized on a solid surface imposes geometric and/or mechanical constrains on filament growth. As actin filaments form part of the cell cortex (*Biro et al., 2013*) and focal adhesions (*Kanchanawong et al., 2010*) where they assemble into oligomeric membrane-anchored complexes with many actin-binding proteins tethered to the plasma membrane surface, our studies may not be too far from the biologically relevant situation in living cells. Moreover, the cell interior is very crowded (*Luby-Phelps, 2000*) and some sub-cellular actin arrays are tightly packed (*Jasnin et al., 2013*). Both of these conditions may lead to the immobilization of actin-binding proteins and generate similar constrains during filament growth. In addition, the presence of side-binding proteins in solution is sufficient for altering the filament kinetics and mechanics. Depending on the local cross-linker protein abundance in the cell, turnover kinetics on the order of 1 μm of filament within ∼1 min can be achieved, a rate at which treadmilling could become a contributing factor to cellular retrograde flow in the lamellipodium (*Watanabe and Mitchison, 2002*; *Ponti et al., 2004*). Additionally, filament structural changes generated by side-binding proteins may also play a more active role in the identity and turnover of actin-based sub-cellular structures than previously thought, by regulating processes such as branching and fragmentation (*Hansen et al., 2013*) or network mechanics (*Jensen et al., 2014*). Given the vast number of side-binding proteins, kinetic modulation via structural alteration may be a general regulatory mechanism of actin dynamics.

## Materials and methods

### Proteins

Actin was obtained from chicken muscle using the method of acetone powder. Actin was extracted by one round of polymerization and pelleting by centrifugation (*Spudich and Watt, 1971*). The resulting pellet was depolymerized in G-buffer (1 mM Tris–HCl pH 7.8, 2 mM ATP, 2 mM $CaCl_2$, 2 mM DTT) overnight at 4°C followed by gel filtration on a Sephacryl S-300 column. Myosin was purified and chemically inactivated with N-Ethyl-Maleimide according to the published protocol (*Breitsprecher et al., 2009*). Atto488-actin, α-actinin, and filamin were purchased from Hypermol (Bielefeld, Germany). Alternatively, actin was labeled with succinimidyl ester atto488 (ATTO-TEC GmbH, Germany) on random lysine residues. Actin labeling was performed under polymerization conditions (50 mM KCl and 2 mM $MgCl_2$) followed by depolymerization and gel filtration in G-buffer. The functionality of 1:1 dye:protein lysine-modified actin was found to be unaffected by the labeling as has been previously characterized using pyrene polymerization assays, TIRF elongation, EM and FCS experiments (*Crevenna et al., 2013*). Unlabeled and labeled actin were mixed to yield a final ratio of 2:1 unlabeled:labeled actin molecules. The actin mixture (20 µl) was snap frozen and stored at −80°C until further use. Before use, an actin aliquot was centrifuged to remove possible aggregates.

A plasmid containing the gene of *Dd* VASP was kindly provided by J Faix, (Hanover, Germany). VASP was expressed using a pCoofy plasmid in Sf9 cells with a MBP-tag and purified following standard methods as described previously (*Scholz et al., 2013*). MBP-VASP was used without cleavage, since removal of the tag resulted in protein aggregation and degradation. For VASP ΔGAB purification, the VASP coding sequence without residues 198–220 was amplified using the pCoofy28-full-length VASP as a template, forward primer 5′-GCGCTTTTATCAACACCGCCACCTGCGGCTGG-3′ and the reverse primer 5′-GCAGGTGGCGGTGTTGATAAAAGCGCTGGTGTACCAACAAAAAC-3′. Then the VASP(delta198-220) coding sequence was further cloned into a pEC-GST vector and expressed using *E. coli* BL21(DE3) as reported previously (*Wang et al., 2014*). Briefly, the *E. coli* strain was grown at 37°C in 2 L of ZY auto-induction medium for 5 hr and then the temperature was reduced to 18°C overnight. Cells were harvested and resuspended in 50 mM Tris pH7.5, 500 mM NaCl, 1 mM DTT supplemented with protease inhibitors and the cells were disrupted using sonication. The protein was purified from clarified cell lysate using a 5-ml GSTrap FF column (GE Healthcare, Germany) with elution buffer 50 mM Tris, pH7.5, 150 mM NaCl, 20 mM reduced glutathione, 1 mM DTT and further purified using size exclusion chromatography (Superdex 75, GE Healthcare, Germany) with buffer 50 mM Tris, pH7.5, 500 mM NaCl, 1 mM DTT.

### Imaging

Flow cells were made as a sandwich of a cover slip (20 × 20 mm), parafilm with an approximate 5-mm wide channel and a glass slide. The surfaces of the flow cells were passivated to avoid adsorption of actin to the sample holder by incubating them with 10% (wt/vol) of BSA in PBS for 10 min. Flow cells were washed three times with 90 µl of G-buffer. The tethering protein was then applied for 5 min and the flow cell was then washed again three times with 90 µl of G-buffer. Actin (33% atto488-actin) was incubated 5 min on ice with 1/10 volume of 10x ME buffer (400 µM $MgCl_2$ and 2 mM EGTA) to exchange $Ca^{2+}$ for $Mg^{2+}$. The actin-containing solution was mixed with imaging buffer (catalase, β-mercaptoethanol, glucose oxidase, 0.8% [vol/vol] D-glucose, 0.25% [wt/vol] methylcellulose, and 1/10 volume of 10x KMEI buffer [500 mM KCl, 20 mM $MgCl_2$, 20 mM EGTA, and 300 mM imidazole], with a final pH of 7.1) and introduced into the flow cell. TIRF microscopy was performed using a TILL photonics inverted microscope (FEI Munich GmbH, Germany). A single actin aliquot was used within 12 hr.

The lattice-binding protein surface density was estimated from the protein concentration, the sample volume (~10 µl) and the surface to which the sample was adsorbed (a flow cell of 5 mm × 20 mm, giving 100 $mm^2$) as done previously (*Howard et al., 1989*; *Crevenna et al., 2008*). All protein in solution was assumed to adsorb on the upper and lower glass surfaces. To achieve consecutive lower tether densities, the total protein concentration was serially diluted. At low tethering protein concentrations, individual filaments swiveled around distinctive attachment points indicating that they are bound to single tethering molecules as observed previously (*Howard et al., 1989*; *Crevenna et al., 2008*). To estimate the density in an alternative manner, we measured the average number of pivot points per micron of filament at the two lowest protein concentrations and divided that by the

average area covered during swiveling. Assuming a linear scaling with protein concentration, this estimation resulted in a slightly lower density (by a factor of 2) compared to those reported in *Figure 3*. The concentration-based estimated densities represent an upper limit and are easy to reproduce. Hence, we report both in the text.

## Two-color solution assay

Filaments were formed using atto565-labeled actin in G-buffer by addition of 1/10 volume of 10x KMEI buffer. After more than 2 hr of polymerization at room temperature, filaments were fragmented by shearing and subsequently mixed with atto488-labeled monomers and allowed to elongate for 15 min. Filaments were then stabilized with unlabeled phalloidin and diluted for imaging on an Epi-Fluorescent Microscope (Axiovert 200, Zeiss, Germany).

## Data analysis

Raw movies were corrected for *x*- and *y*- stage drift by first calculating its magnitude via image correlation spectroscopy (*Hebert et al., 2005*), and secondly, correcting the drift by bicubic interpolation. Drift estimation and correction were implemented in custom programs written in LabView and MATLAB (The MathWorks, MA). Kymographs of single filaments were made using Metamorph or Image J, while further analysis was carried out using MATLAB. Filament analysis tools are available at: http://www.cup.uni-muenchen.de/pc/lamb/actin_filament_dynamics.html. The position of the filament tip, per line in the kymograph, was estimated by fitting an error function as previously described (*Demchouk et al., 2011*). More than 20 filaments were analyzed per condition. To estimate the first pause distribution, we used the model described by *Niedermayer et al. (2012)* with $\omega = 2 \times 10^6$. The light intensity for treadmilling experiments ranged from 0.74 to 0.92 mW·mm$^{-2}$. Growth orientation was assessed manually with the following criteria: Barbed-end filament growth direction was classified as straight/not-defined, clockwise or counterclockwise from experiments at the highest surface tether density.

## Monte Carlo simulations

For the model presented in *Figure 3*, we used a Monte Carlo method to simulate the polymerization of actin filaments at the barbed-end. For each condition, a $10^5$ monomer long actin filament was polymerized, and the instantaneous elongation rate was calculated for every point and then averaged over the total length of the filament. The average elongation rate was calculated from the length of the polymer over time for each condition. The effect of the tethering protein was simulated by a change in the effective $k_{on}$ at the site and vicinity of the tethering protein (*Figure 3—figure supplement 4*). Over the $10^5$ monomer sites, $N_b$ side-binding proteins were randomly placed corresponding to the desired side-binding protein density. This gives on the order of $10^4$ side binding proteins for the highest densities computed. First, the tethering protein's positions were randomly chosen according to the protein density. The surface density was calculated assuming that all added tethering proteins adsorbed to the surface and were functionally active. To convert from surface density to fractional occupancy, we used the area occupied by 1 μm of actin filament (0.006 μm$^2$), using a value of 370 subunits per micron of filament. We neglected tether dissociation from the filament, as this would only reduce the effective tether density. During the simulated polymerization, the effective $k_{on}$ was changed to $\alpha \cdot k_{on}^0$ at the position of the tethering protein and decreased linearly until reaching the free actin value of $k_{on}^0$ after a characteristic length ($L_C$) counted in monomers of actin. The values of 11 μM$^{-1}$s$^{-1}$ and 2 s$^{-1}$ for barbed-end $k_{on}^0$ and $k_{off}$, respectively, were taken from literature (*Pollard, 1986*), and the average elongation rate was taken from measurements at the lowest tethering protein density (*Figure 3E*). A first round of simulations was performed to roughly estimate the optimal interval for the parameters ($\alpha$ and $L_C$). A second set of simulations over restrained intervals, with a better resolution on $\alpha$ and $L_C$, yielded a 3D space ($\alpha$, $L_C$, v). The 2D $\chi^2$ was calculated, the minimum value gave the best ($\alpha$, $L_C$) and the confidence intervals were taken by $\Delta\chi^2 = 1$ (68%) and $\Delta\chi^2 = 4$ (95%). The only free parameters were $\alpha$ and $L_C$, which were determined for each curve (i.e., elongation as a function of lattice-binding protein density) by comparing the simulated elongation rates to the experimental values and minimizing the $\chi^2$ using:

$$\chi^2(\alpha, L_C) = \sum_{i=1}^{N} \left(v(x_i) - v_{sim}(x_i, \alpha, L_C)/std(x_i)\right)^2,$$

where $v(x_i)$ is the experimental velocity for density $i$, $v_{sim}(x_i, \alpha, L_C)$ is the corresponding simulated velocity with parameters $\alpha$ and $L_C$, and $std(x_i)$ is the experimental standard deviation for this data point. Hence, the only free parameters are $\alpha$ and $L_C$ and these were determined for each curve (i.e., elongation as a function of lattice-binding protein density). All simulations were done in MATLAB.

### Persistence length calculation

Individual filaments were extracted from the measured data using the algorithm of 'open active contours' within JFilamin, a plug in for Image J (*Li et al., 2009*; *Smith et al., 2010*). The filament persistence length '$L_p$' was determined by calculating the angular correlation (*Isambert et al., 1995*):

$$\langle cos[\theta(s) - \theta(0)]\rangle = e^{-s/2L_p},$$

where the brackets represent the average correlation function of the tangent $\theta$, measured along the contour length $s$. The point spacing used to reconstruct a single filament was between 6 and 10 points per micron to avoid artifacts in the $L_p$ estimation (*Isambert et al., 1995*; *McCullough et al., 2008*; *Smith et al., 2010*). All data analysis was done in MATLAB.

### Local change in actin concentration model

The strong increase in the elongation rate in the presence of, for example, VASP as a tethering protein was first thought to originate from the multiple actin monomer binding sites on each VASP protein (*Breitsprecher et al., 2008*, *Hansen et al., 2010*, *Breitsprecher et al., 2011*). Theoretically, the polymerization kinetics is expected to be inhomogeneous along the actin filament and only to be locally enhanced through a higher local concentration of free actin monomers due to the presence of VASP. The growth at one end of the polymer can be written as follows:

$$E(x) = k_{on}c(x) - k_{off}, \tag{1}$$

where $E(x)$ is the elongation rate at the position $x$ along the filament axis, $c(x)$ the local concentration of globular actin at this position, and $k_{on}$ and $k_{off}$ are the association and dissociation rates. The average elongation rate along the filament length is given by,

$$\langle E\rangle = k_{on}\langle c\rangle - k_{off}, \tag{2}$$

where $\langle c\rangle = c_0(1+4d)$, $c_0$ is the free actin concentration in solution and $d$ is the density of VASP protein at the surface. Given that one VASP protein can bind 4 actin monomers, the local concentration of actin monomers available for polymerization can be significantly increased at the site of a tethering protein. The relationship between the average elongation rate and the protein surface density was expected to be linear as follows:

$$\langle E\rangle = k_{on}c_0(1+4d) - k_{off}. \tag{3}$$

Experimental points do not show a linear dependency on the protein surface density as expected from this model (*Figure 3—figure supplement 5*). An alternative mode of operation for VASP has been recently postulated where the protein not only increases the local concentration but also transfers monomers from its monomer binding domains to the filament tip (*Breitsprecher et al., 2011*). The surface density dependency of this alternative model would, nonetheless, predict a linear behavior as well, albeit with a different slope. In addition, this increased local concentration model would not explain the effect observed when using filamin, α-actinin, or with the VASP ΔGAB construct, which is unable to bind monomers.

## Acknowledgements

We thank D Kovar for the Image J macros's to perform the filament analysis, J Faix for the DdVASP expression plasmid and J Dominguez for valuable discussions. This work was supported by the Max Planck Society and by grants from the Deutsche Forschungsgemeinschaft through the SFB 1035 (to OFL), the SPP 1464 (to RWS and DCL), the Excellent Clusters Nanosystems Initative Munich (NIM) and Center for Integrated Protein Science Munich (CIPSM), the Cells-in-Motion Cluster of Excellence (EXC1003 – CiM) of the University of Münster, a CONACYT-DAAD scholarship A/09/7253 (to MA) and the Ludwig-Maximilians-Universität München (via the LMUInnovativ BioImaging Network (BIN) and the Center for NanoScience (CeNS)). This research has also been supported (to KK) by the 2011/01/B/NZ1/00031 grant from the National Science Centre, Poland.

# Additional information

## Funding

| Funder | Grant reference | Author |
| --- | --- | --- |
| Deutsche Forschungsgemeinschaft | SPP 1464 | Roland Wedlich-Söldner, Don C Lamb |
| Deutsche Forschungsgemeinschaft | SFB 1035 | Oliver F Lange |
| Consejo Nacional de Ciencia y Tecnología | CONACYT DAAD-A/09/7253 | Marcelino Arciniega |
| Narodowe Centrum Nauki | 2011/01/B/NZ1/00031 | Kaja Kowalska |

The funders had no role in study design, data collection and interpretation, or the decision to submit the work for publication.

## Author contributions

AHC, Conception and design, Acquisition of data, Analysis and interpretation of data, Drafting or revising the article; MA, Design and performed all Brownian dynamics computer simulations; AD, Design and analyzed Monte Carlo model and simulations, Analysis and interpretation of data, Drafting or revising the article; NM, Contributed reagents, Analysis and interpretation of data, Drafting or revising the article; KK, Acquisition of data, Contributed unpublished essential data or reagents; OFL, RW-S, Analysis and interpretation of data, Drafting or revising the article, Contributed unpublished essential data or reagents; DCL, Conception and design, Analysis and interpretation of data, Drafting or revising the article

## Author ORCIDs

Alvaro H Crevenna, http://orcid.org/0000-0002-5580-3711

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
