## [Decision Letter]

[Editors’ note: this article was originally rejected after discussions between the reviewers, but the authors were invited to resubmit after an appeal against the decision.]

Thank you for choosing to send your work entitled “Side-binding proteins modulate actin filament dynamics” for consideration at *eLife*. Your full submission has been evaluated by Vivek Malhotra (Senior editor) and 3 peer reviewers, one of whom is a member of our Board of Reviewing Editors, and the decision was reached after discussions between the reviewers. Based on our discussions and the individual reviews below, we regret to inform you that your work will not be considered further for publication in *eLife*.

As you can see from the reviews, there is enthusiasm for the role of side binding in modulating actin filament dynamics. There is no question in our minds that this could be a significant contribution to the field. However, we feel that there is a lot of work required to make many of the experiments solid.

We think that one of the key problems is summarized by Reviewer 1. Your main results come from VASP, while the other proteins have a more modest effect.

The problem with VASP as a 'side-binder' is that it contains a g-actin binding domain and can function as a polymerase. We think at minimum you would need to perform experiments using VASP with a debilitated or missing g-actin binding domain to unambiguously ascribe the effects they observed to the side-binding activity of VASP.

A second problem highlighted by another reviewer is that the experiments here were all done with the side-binding proteins acting to tether the actin to the slide surface. It seems important to also conduct experiments where the side-binders are not tethered. It is possible that some of the effects seen here result from torque induced by sequentially attached immobilized proteins, at least at high tethering density, and also the kinetic of side-binder binding and detachment.

Reviewer #1

In this paper, the authors report the effects of 'side-binding' proteins on actin polymerization kinetics. They use TIRF microscopy to measure the rates of growing and shrinking at the level of individual actin filaments, observe that the side-binders give different density-dependent effects, and attempt to connect their findings to the structure of the filament using computational simulations. This is a potentially interesting study that combines a variety of approaches to study a challenging and timely problem: how allostery can modulate the polymerization dynamics of biologically important polymers. From my perspective the most interesting claim is that different side-binding proteins can differentially tune the polymerization kinetics by perturbing/modulating the structure of the filament over length scales that exceed nearest neighbor relationships. Even while recognizing the challenging and interesting nature of the subject under study, however, I am not convinced that the authors' data provide unambiguous support for their headline claims.

Major concerns:

1) VASP gives by far the largest effects (e.g. 10-fold increase in apparent *k*_on_, etc.), and is presented and interpreted as a 'side-binding' protein. Without the big effect evident from VASP, I think the results are less striking because much more modest in nature. The problem with VASP as a 'side-binder' is that it contains a g-actin binding domain and can function as a polymerase. I think at minimum the authors would need to perform experiments using VASP with a debilitated or missing g-actin binding domain to unambiguously ascribe the effects they observed to the side-binding activity of VASP. Otherwise they need to do some significant re-phrasing and acknowledge that maybe the VASP polymerase activity is contributing to the fast elongation, and this will mitigate and confuse the impact of their findings about the possible role(s) of side-binders. I think such an experiment should be readily achievable.

2) Excepting VASP, the observed effects on instantaneous growth rates are relatively modest and require substantial amounts of the side-binder. Effects on pausing are more substantial but it seems possible those are artifacts (see next point). In any event, pausing does not seem to be the core point of the paper. If I understand Figure 3 properly, beside VASP there is little effect on 'pause-free E' until a density of about 1000 side-binders per μM^2. If I calculated right that is 1 side-binder roughly every 8 nm. But at 370 actins/μM of polymer, 8 nm average spacing between side-binders equates to something like one every 3 actins. Requiring such high densities undermines the claim of longer-range effects.

3) The authors presented some controls for non-specific surface effects, but I'm not sure they can rule out specific surface effects. Is the rate of filament growing or shrinking affected by how close the filament end is to the coverslip? Is it possible to do a subset of these experiments with side-binder in the solution but not on the coverslip (and using low densities of NEM-myosin on the coverslip)? If so, and if similar changes were observed it would help argue against the possibility that the results were an artifact of having a coverslip coated with filament binders. I don't view such an experiment as essential, but it would help. Likewise, unless I am overlooking something the bending seemed like it could be largely explained by a model in which the high density of side-binder simply 'grabbed' a fluctuating filament end.

4) There seems to be too wide a gap between the TIRF measurements and the Brownian dynamics simulations. The simulations show that different filament structures can give rise to different on rate constants, but by itself this claim is not surprising to me and I don't think it can be taken as evidence that the side-binders actually exerted their effects by altering the structure of the filament. Maybe they did, of course, but alternative explanations have not been convincingly ruled out. How to fix this is not obvious, but for me the connection between the TIRF results and the Brownian dynamics was too tenuous, and for me this weak connection undermined the thrust of their interpretation.

Reviewer #2

The relationship between actin filament structure and actin filament dynamics, and how these can be modulated by filament binding proteins, is a long-standing and important question in the cytoskeletal field. Although some studies suggest that side-binding proteins can, through long range conformational effects, affect the kinetics and thermodynamics of monomer addition and dissociation at the filament ends, this has not yet been carefully examined. Moreover, there is some controversy in the recent literature on single actin filament dynamics arising from technical problems in the measurements. The work here addresses these issues through quantitative TIRF imaging of filament dynamics, and some computational analyses. The authors have used a new, more accurate procedure to determine the instantaneous position of the filament ends during imaging, which has revealed new aspects of filament dynamics not observed (or observable) in previous studies. This has led to several interesting conclusions, including that different side-binding proteins have different effects on actin binding and dissociation rates at the filament ends, and that the pointed end significantly populates a kinetically-trapped state.

In my opinion these advances would merit publication in *eLife*. However, I have concerns about some of the data that the authors will need to address before the work is technically solid. I am also less positive about the computational work at the end of the manuscript, and feel that is detracts from the experimental observations and should be down-played or removed.

First, the experiments here were all done with the side-binding proteins acting to tether the actin to the slide surface. It seems important to also conduct experiments where the side-binders are not tethered. It is possible that some of the effects seen here result from torque induced by sequentially attached immobilized proteins, at least at high tethering density, and also the kinetic of side-binder binding and detachment. Perhaps one molecule could be used at low density to immobilize the filaments, and a second could be added in solution?

Second, it is unclear to me how the kinetic parameters at the pointed end measured here for single filaments agree so well with previous bulk measurements, given the high pause probability illustrated in Figure 4. That is, according to line 201, the *k*_on_ and *k*_off_ values in Table 1 are derived from the pause-free elongation rates. But the previously reported bulk measurements should represent a convolution of those parameters with the pausing probability, which is very high for the pointed end (Figure 4). So the bulk rate constants should be much lower than those measured from the pause-free portions of the single-filament data.

Third, given recent controversies regarding the effects of unintended photo-crosslinking on actin filament dynamics (46; 35), the authors need to make certain that the interesting behaviors they observe are independent of such processes. The data in the Figure 4 supplements are compelling in this regard. Nevertheless, both of their fluorophores, atto565 and atto488, are attached to actin in the same manner, through non-specific coupling to lysine residues. It is possible that the modified sites are particularly prone to photo-crosslinking. It would add even greater strength to their arguments (and be easy to do) to also try maleimide labeling, which uniquely modifies cysteine 376 in actin. Related to this point, the authors need to characterize and describe the labeling of their actin with atto488/565: how many sites on average were modified, what was the variability from preparation to preparation, what collection of sites were modified? How do they know that the labeling did not affect the properties of the actin?

Fourth, the modeling to fit the data in Figure 3 should be described in more detail in the main text. This information should not all be shifted to the methods section. Why does the effect of VASP propagate over a much longer range than the other ligands? Shouldn't *L*_*C*_ be largely dependent on actin itself? It's hard to see how there could be 2 orders of magnitude difference between α-actinin and VASP. Can the authors explain this based on other observations? Perhaps related, while I am not an expert at modeling, the huge 68% confidence interval and large difference between VASP and the other proteins raises concerns about this analysis.

Finally, in my opinion the computations at the end of the paper do not strengthen the conclusions of the work, and provide a false sense of validation of the experimental data. I am highly suspect of the ability of MD simulations to accurately predict binding rates from the different structural models. The system is too complex, and the structural models themselves are at too low resolution. Further, it is almost certain that an actual filament samples many different structures contained within the manifold sampled by the available models. The finding that different structures produce different predicted rates in the calculations is not surprising; different starting points should give different simulations. But I don't see how these computed rates can be meaningfully interpreted or related back to the experimental data (e.g. is the Namba mode 3 structure a better model for the filament because it gives kinetic parameters that better match the experimental data? We can't know, since we don't know how well the simulations predict the kinetics in the first place). So I don't feel that the simulations either support or refute a relationship between structure and kinetics.

Reviewer #3

Crevenna et al. used TIRF microscopy to conduct a detailed study on how the dynamics at the barbed end and the pointed end of the actin filaments change in response to binding of four proteins to the side of actin filaments. As summarized in Table 1, the binding of these proteins often changed the *k*_on_ and *k*_off_ at the barbed and pointed ends significantly. Therefore, this reviewer agrees with the authors that binding of proteins to the side of actin filaments “may be a ubiquitous mechanism to generate the rich variety of observed cellular actin dynamics”. This reviewer finds this well documented here and interesting.

The authors used Monte-Carlo simulations to understand why side binding proteins change dynamics at the ends of actin filaments. They highlight shortcomings of models proposed earlier, and propose that these protein work through “exploiting the natural malleability of the actin filament structure”. This reviewer finds the proposal of the authors reasonable. However, the precise mechanism of how the side-binding proteins affect dynamics at the ends of actin filaments remains unclear.

Major concerns:

1) The authors document a “kinetically inactive phase” at the pointed end of actin filaments. This reviewer finds this interesting. The authors also report high pausing probability in presence of NEM-myosin (Figure 3). Since NEM-myosin has been widely used in TIRF assays to study dynamics of individual actin filaments, the author should comment if kinetically inactive pointed ends and paused barbed ends have been observed before, and how these measurements compare to those reported in this study. In case these have not been observed before, what did the authors do differently from previous studies?

2) The authors document that at higher side-binding protein density, actin filaments tend to grow either in a clockwise or in a counter-clockwise fashion. In presence of α-actinin, the actin filaments grow almost exclusively in a counter-clockwise fashion (Figure 5). The authors should repeat these measurements for long actin filaments to test if their conclusion still holds. For instance, see the micrograph in α-actinin case in Figure 5.

---

## [Author Response]

We were very pleased that all three reviewers were very positive regarding our research and that they considered it to have the potential to be a significant contribution to the field. From your decision letter, we understand that “It is the policy of eLife not to ask for substantial experimental work before acceptance.” The recommended experiments are feasible within a short time (90 days). Hence, we do not find the additional experimental work to be “substantial”. In addition, they would strengthen our findings and provide more convincing evidence of our proposed interpretation. Therefore, we would like to ask you to consider a revised version of our manuscript that incorporates the reviewers’ comments and the additional suggested work.

*The problem with VASP as a 'side-binder' is that it contains a g-actin binding domain and can function as a polymerase. We think at minimum you would need to perform experiments using VASP with a debilitated or missing g-actin binding domain to unambiguously ascribe the effects they observed to the side-binding activity of VASP*.

*A second problem highlighted by another reviewer is that the experiments here were all done with the side-binding proteins acting to tether the actin to the slide surface. It seems important to also conduct experiments where the side-binders are not tethered. It is possible that some of the effects seen here result from torque induced by sequentially attached immobilized proteins, at least at high tethering density, and also the kinetic of side-binder binding and detachment*.

We thank the reviewers and the editors for the constructive comments. To address the two main issues summarized by the editor above, we have performed measurements with VASP-ΔGAB and with a low density of tethering proteins and additional side-binding proteins in solution. The results have been included in the revised manuscript. Below is the detailed point-by-point response to the comments raised by the reviewers.

Reviewer #1

*In this paper, the authors report the effects of 'side-binding' proteins on actin polymerization kinetics. They use TIRF microscopy to measure the rates of growing and shrinking at the level of individual actin filaments, observe that the side-binders give different density-dependent effects, and attempt to connect their findings to the structure of the filament using computational simulations. This is a potentially interesting study that combines a variety of approaches to study a challenging and timely problem: how allostery can modulate the polymerization dynamics of biologically important polymers. From my perspective the most interesting claim is that different side-binding proteins can differentially tune the polymerization kinetics by perturbing/modulating the structure of the filament over length scales that exceed nearest neighbor relationships. Even while recognizing the challenging and interesting nature of the subject under study, however, I am not convinced that the authors' data provide unambiguous support for their headline claims*.

*Major concerns*:

*1) VASP gives by far the largest effects (e.g. 10-fold increase in apparent* k_*on*_*, etc.), and is presented and interpreted as a 'side-binding' protein. Without the big effect evident from VASP, I think the results are less striking because much more modest in nature. The problem with VASP as a 'side-binder' is that it contains a g-actin binding domain and can function as a polymerase. I think at minimum the authors would need to perform experiments using VASP with a debilitated or missing g-actin binding domain to unambiguously ascribe the effects they observed to the side-binding activity of VASP. Otherwise they need to do some significant re-phrasing and acknowledge that maybe the VASP polymerase activity is contributing to the fast elongation, and this will mitigate and confuse the impact of their findings about the possible role(s) of side-binders. I think such an experiment should be readily achievable.*

We thank the reviewer for this excellent suggestion and have now included experiments using a VASP construct with a deletion of the GAB domain. Filaments growing with this VASP ΔGAB construct as a tether showed very similar behavior to those using the full length VASP construct. The new data has now been incorporated throughout the manuscript and strengthens the hypothesis that it is the side-binding capacity of VASP that alters kinetics of filaments.

*2) Excepting VASP, the observed effects on instantaneous growth rates are relatively modest and require substantial amounts of the side-binder. Effects on pausing are more substantial but it seems possible those are artifacts (see next point). In any event, pausing does not seem to be the core point of the paper. If I understand*
Figure 3
*properly, beside VASP there is little effect on 'pause-free E' until a density of about 1000 side-binders per μM^2. If I calculated right that is 1 side-binder roughly every 8 nm. But at 370 actins/μM of polymer, 8 nm average spacing between side-binders equates to something like one every 3 actins. Requiring such high densities undermines the claim of longer-range effects*.

We apologize for this confusion. To convert from surface density to filament density, we have multiplied the surface density with the area occupied by 1 micron of filament (1μm x 6nm). Using this calculation, (1000 molecules μm^-2^ * 0.006 μm^2^ filament area) we obtain that VASP starts to exert an effect in kinetics at binding densities of ∼ 6 molecules per micron of filament and at filament densities of 10-20 binders per μm for the other proteins, supporting our observation of long-ranged interactions. To aid in the result presentation and help the reader in understanding the effects of side-binders on the filament growth, we have added a second x-axis in Figure 3 and we now mention the linear density throughout the manuscript.

*3) The authors presented some controls for non-specific surface effects, but I'm not sure they can rule out specific surface effects. Is the rate of filament growing or shrinking affected by how close the filament end is to the coverslip? Is it possible to do a subset of these experiments with side-binder in the solution but not on the coverslip (and using low densities of NEM-myosin on the coverslip)? If so, and if similar changes were observed it would help argue against the possibility that the results were an artifact of having a coverslip coated with filament binders. I don't view such an experiment as essential, but it would help. Likewise, unless I am overlooking something the bending seemed like it could be largely explained by a model in which the high density of side-binder simply 'grabbed' a fluctuating filament end*.

Again, we thank the reviewer for this excellent suggestion and agree that experiments with a low density of NEM-myosin on the surface while another side-binding protein is present in solution would help in clarify the origin of pauses. Therefore, we performed experiments with VASP, VASP ΔGAB or filamin in solution while filaments are tether using a low density of NEM-myosin. The growth behavior of such filaments was similar to what we observed when the side-binding protein was used as the tethering protein. These results are now discussed throughout the main text and included in Figures 1, 2, 3 and 5.

In Figure 5, we have also plotted the persistent length for filaments at low tethering densities while having the side-binding proteins in solution. In these experiments, a significant change in persistent length is observable although slightly smaller than for experiments performed at high tethering protein densities. Hence, we can conclude that bending is due to binding of the side-binding proteins and not a surface-induced affect.

*4) There seems to be too wide a gap between the TIRF measurements and the Brownian dynamics simulations. The simulations show that different filament structures can give rise to different on rate constants, but by itself this claim is not surprising to me and I don't think it can be taken as evidence that the side-binders actually exerted their effects by altering the structure of the filament. Maybe they did, of course, but alternative explanations have not been convincingly ruled out. How to fix this is not obvious, but for me the connection between the TIRF results and the Brownian dynamics was too tenuous, and for me this weak connection undermined the thrust of their interpretation*.

We carried out Brownian dynamics simulations to explore the idea that subtle structural changes could give rise to changes in kinetics, which we observed in our estimates of the kinetic rates based on the different filament models. However, we acknowledge that there is little information regarding how the side-binding proteins modulate the filament structure and, more importantly, the connection is weak on how this structural alteration might translate into a kinetic effect. Therefore, we have now removed the entire section regarding the Brownian dynamics simulations from the manuscript.

Reviewer #2

*The relationship between actin filament structure and actin filament dynamics, and how these can be modulated by filament binding proteins, is a long-standing and important question in the cytoskeletal field. Although some studies suggest that side-binding proteins can, through long range conformational effects, affect the kinetics and thermodynamics of monomer addition and dissociation at the filament ends, this has not yet been carefully examined. Moreover, there is some controversy in the recent literature on single actin filament dynamics arising from technical problems in the measurements. The work here addresses these issues through quantitative TIRF imaging of filament dynamics, and some computational analyses. The authors have used a new, more accurate procedure to determine the instantaneous position of the filament ends during imaging, which has revealed new aspects of filament dynamics not observed (or observable) in previous studies. This has led to several interesting conclusions, including that different side-binding proteins have different effects on actin binding and dissociation rates at the filament ends, and that the pointed end significantly populates a kinetically-trapped state*.

*In my opinion these advances would merit publication in* eLife*. However, I have concerns about some of the data that the authors will need to address before the work is technically solid. I am also less positive about the computational work at the end of the manuscript, and feel that is detracts from the experimental observations and should be down-played or removed.*

First, the experiments here were all done with the side-binding proteins acting to tether the actin to the slide surface. It seems important to also conduct experiments where the side-binders are not tethered. It is possible that some of the effects seen here result from torque induced by sequentially attached immobilized proteins, at least at high tethering density, and also the kinetic of side-binder binding and detachment. Perhaps one molecule could be used at low density to immobilize the filaments, and a second could be added in solution?

This is the same important issue that has been raised by Reviewer 1 and we agree with the reviewers that experiments with a low density of tethering molecules and a second binding molecule in solution would help to rule out torque-induced effects. Therefore, we have performed experiments with VASP, VASP ΔGAB or filamin in solution while filaments are tether using a low density of NEM-myosin. We observed a similar growth behavior of such filaments to that when the side-binding protein was used as a tether. These results are now discussed throughout the main text and included in Figure 3—figure supplement 1, Figure 3—figure supplement 2 and Figure 3—figure supplement 3 and Figure 5.

*Second, it is unclear to me how the kinetic parameters at the pointed end measured here for single filaments agree so well with previous bulk measurements, given the high pause probability illustrated in*
Figure 4*. That is, according to line 201, the* k_*on*_
*and* k_*off*_
*values in*
Table 1
*are derived from the pause-free elongation rates. But the previously reported bulk measurements should represent a convolution of those parameters with the pausing probability, which is very high for the pointed end (*Figure 4*). So the bulk rate constants should be much lower than those measured from the pause-free portions of the single-filament data.*

We apologize for the confusion on this important point and we have now clarified this in the text. Our pause-free elongation rates convoluted with observed pause rate (∼90%) would result in elongation rates (*k*_on_ of ∼ 0.2 and *k*_off_ ∼0.1) consistent with previously estimated rates using TIRF microscopy (Kuhn and Pollard, Biophys J, 2001, Table 1). The pause-free elongation rates agree with data obtained from EM data (Pollard, JCB, 1986) (misnamed “bulk” in the manuscript). EM data was obtained by measuring the length of individual filaments at various times points from 20 to 60 s using negative-stain electron microscopy (Pollard, JCB, 1986). We have noticed that pointed-end growth also occurs at the beginning of filament growth (Figure 4) before going to a paused state. One possible explanation for this difference is that the “initial pointed-end growth” could originate from an unconstrained ATP-actin monomer-based elongation. The appearance of the paused state could originate from ATP hydrolysis, actin flattening and/or a non-elongating structure (Narita et al., EMBO J, 2010) at the pointed-end. The origin of the paused state and it possible relationship to nucleotide or structure is unclear at the moment and we are working towards elucidating the mechanism behind it. We have included this discussion in the main text (subsection headed “Intrinsic filament dynamics”).

*Third, given recent controversies regarding the effects of unintended photo-crosslinking on actin filament dynamics (*[46]*;*
[35]*), the authors need to make certain that the interesting behaviors they observe are independent of such processes. The data in the*
Figure 4
*supplements are compelling in this regard. Nevertheless, both of their fluorophores, atto565 and atto488, are attached to actin in the same manner, through non-specific coupling to lysine residues. It is possible that the modified sites are particularly prone to photo-crosslinking. It would add even greater strength to their arguments (and be easy to do) to also try maleimide labeling, which uniquely modifies cysteine 376 in actin. Related to this point, the authors need to characterize and describe the labeling of their actin with atto488/565*: *how many sites on average were modified, what was the variability from preparation to preparation, what collection of sites were modified? How do they know that the labeling did not affect the properties of the actin?*

To ensure actin functionality is indeed a key issue with respect to protein labeling. We agree with the reviewer that, whenever possible, it is preferable to use a specific labeling strategy than stochastic labeling. Unfortunately, the use of the cysteine residue at position 376 for fluorophore attachment is problematic as the polymerizability is compromised (Kuhn and Pollard, Biophys J, 2001) and elongation rates are decreased with increasing amounts of labeled protein. In order to test for possible artifacts with stochastic labeling, we have previously characterized the functional properties of lysine atto-labeled actins where we used 1:1 dye:protein ratios for labeling (Crevenna et al., JBC, 2013), which leads to an average labeling of one fluorophore per protein. In Crevena et al., JBC, 2013, we carried out pyrene polymerization assays, TIRF elongation, EM and FCS experiments. Our results suggested that the lysine-labeled protein is fully functional. We have not observed (by mass spectrometry) a preference for lysine modification of the solvent exposed residues (data not shown). For TIRF assays, we mixed fully-functional labeled actin (Crevenna et al., JBC, 2013) with unlabeled actin to a final ratio of 2:1 unlabeled:labeled protein. We have now added the requested detailed information regarding the labeling procedure, etc., in the Material and methods.

*Fourth, the modeling to fit the data in*
Figure 3
*should be described in more detail in the main text. This information should not all be shifted to the methods section. Why does the effect of VASP propagate over a much longer range than the other ligands? Shouldn't* L_C_
*be largely dependent on actin itself? It's hard to see how there could be 2 orders of magnitude difference between α-actinin and VASP. Can the authors explain this based on other observations? Perhaps related, while I am not an expert at modeling, the huge 68% confidence interval and large difference between VASP and the other proteins raises concerns about this analysis.*

We have now included the model section as part of the main text and expanded the text to describe the model in more detail.

The *L*_*C*_ is dependent on the properties of the actin filament itself but is clearly modulated by the side-binding protein. The propagation length of the VASP ΔGAB construct is roughly 2 helical repeats, VASP is 4 helical repeats and the effects of α-actinin, filamin and NEM-myosin are on the order of 13 monomers, i.e. half a helical repeat. This could possibly implicate that the structure of the actin is important in determining the propagation length. However, as this is very speculative, we have decided not to discuss this in the current manuscript.

To have an estimate of the propagation length, we proposed the simplest model necessary. Hence, we have two free parameters (an amplitude and the *L*_*C*_) and wished to avoid making more assumptions (e.g. fixing the amplitude). The results of our model yield the above range in *L*_*C*_ values. However, the important point here is that altered kinetics parameters can be propagated along the actin filament in a side-binding protein dependent manner.

*Finally, in my opinion the computations at the end of the paper do not strengthen the conclusions of the work, and provide a false sense of validation of the experimental data. I am highly suspect of the ability of MD simulations to accurately predict binding rates from the different structural models. The system is too complex, and the structural models themselves are at too low resolution. Further, it is almost certain that an actual filament samples many different structures contained within the manifold sampled by the available models. The finding that different structures produce different predicted rates in the calculations is not surprising; different starting points should give different simulations. But I don't see how these computed rates can be meaningfully interpreted or related back to the experimental data (e.g. is the Namba mode 3 structure a better model for the filament because it gives kinetic parameters that better match the experimental data? We can't know, since we don't know how well the simulations predict the kinetics in the first place). So I don't feel that the simulations either support or refute a relationship between structure and kinetics*.

As we mentioned to Reviewer 1, we carried out Brownian dynamics simulations to explore the idea that subtle structural changes could give rise to changes in kinetics, which we do observe in our estimates based on the different filament models. However, we do acknowledge that there is little information about how the side-binding proteins modulate the filament structure and, more importantly, the connection is weak on how this structural alteration might translate into a kinetic effect. Therefore, we have now removed the whole section regarding Brownian dynamics simulations from the manuscript.

Reviewer #3

*Crevenna et al. used TIRF microscopy to conduct a detailed study on how the dynamics at the barbed end and the pointed end of the actin filaments change in response to binding of four proteins to the side of actin filaments. As summarized in*
Table 1*, the binding of these proteins often changed the* k_*on*_
*and* k_*off*_
*at the barbed and pointed ends significantly. Therefore, this reviewer agrees with the authors that binding of proteins to the side of actin filaments “may be a ubiquitous mechanism to generate the rich variety of observed cellular actin dynamics”. This reviewer finds this well documented here and interesting.*

*The authors used Monte-Carlo simulations to understand why side binding proteins change dynamics at the ends of actin filaments. They highlight shortcomings of models proposed earlier, and propose that these protein work through “exploiting the natural malleability of the actin filament structure”. This reviewer finds the proposal of the authors reasonable. However, the precise mechanism of how the side-binding proteins affect dynamics at the ends of actin filaments remains unclear*.

*Major concerns*:

*1) The authors document a “kinetically inactive phase” at the pointed end of actin filaments. This reviewer finds this interesting. The authors also report high pausing probability in presence of NEM-myosin (*Figure 3*). Since NEM-myosin has been widely used in TIRF assays to study dynamics of individual actin filaments, the author should comment if kinetically inactive pointed ends and paused barbed ends have been observed before, and how these measurements compare to those reported in this study. In case these have not been observed before*, *what did the authors do differently from previous studies?*

Kinetic pauses (distinct from depolymerization pauses due to photodimerization) have been observed previously in the literature. However, they have been considered as potential artifacts and not as an intrinsic property of filaments. We have rephrased the main text to clarify this point. The text now reads: “Although growth pauses have been previously observed during filament elongation measured using total internal reflection fluorescence (TIRF) microscopy (Kuhn et al., 2005; [18]), these pauses were attributed to artifacts and were not characterized further.)

*2) The authors document that at higher side-binding protein density, actin filaments tend to grow either in a clockwise or in a counter-clockwise fashion. In presence of α-actinin, the actin filaments grow almost exclusively in a counter-clockwise fashion (*Figure 5*). The authors should repeat these measurements for long actin filaments to test if their conclusion still holds. For instance, see the micrograph in α-actinin case in*
Figure 5.

We have analyzed our data as a function of filament length. We observe the counter-clockwise growth for filaments of all lengths (see Figure 6 below). In the main text (Figure 5), the percentage of counter-clock growth has been estimated taking all lengths into consideration. That this result is independent of filament length has now been added to the main text lines (in the subsection “Structural effects of side-binding proteins on filaments”).

Author response image 1.**DOI:**
http://dx.doi.org/10.7554/eLife.04599.021